# Learning Distributions over Permutations and Rankings with Factorized Representations

**Daniel Severo**
FAIR at Meta
dsevero@meta.com

**Brian Karrer**
FAIR at Meta
briankarrer@meta.com

**Niklas Nolte**
FAIR at Meta
nolte@meta.com

## Abstract

Learning distributions over permutations is a fundamental problem in machine learning, with applications in ranking, combinatorial optimization, structured prediction, and data association. Existing methods rely on mixtures of parametric families or neural networks with expensive variational inference procedures. In this work, we propose a novel approach that leverages alternative representations for permutations, including Lehmer codes, Fisher-Yates draws, and Insertion-Vectors. These representations form a bijection with the symmetric group, allowing for unconstrained learning using conventional deep learning techniques, and can represent any probability distribution over permutations. Our approach enables a trade-off between expressivity of the model family and computational requirements. In the least expressive and most computationally efficient case, our method subsumes previous families of well established probabilistic models over permutations, including Mallow's and the Repeated Insertion Model. Experiments indicate our method significantly outperforms current approaches on the jigsaw puzzle benchmark, a common task for permutation learning. However, we argue this benchmark is limited in its ability to assess learning probability distributions, as the target is a delta distribution (i.e., a single correct solution exists). We therefore propose two additional benchmarks: learning cyclic permutations and re-ranking movies based on user preference. We show that our method learns non-trivial distributions even in the least expressive mode, while traditional models fail to even generate valid permutations in this setting.

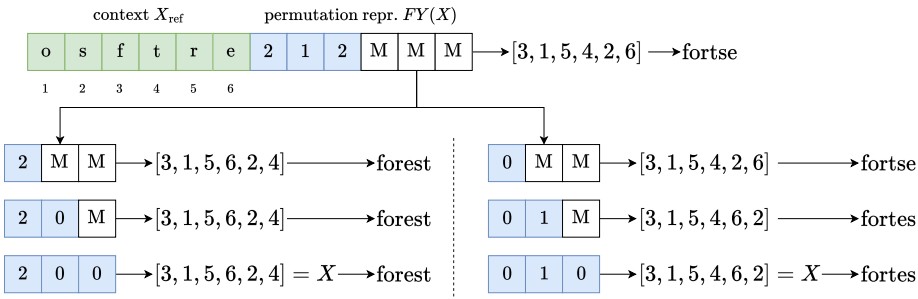

Figure 1: Overview of our method unscrambling the sequence "osftre" autoregressively using one of the representations we consider in this work: Fisher-Yates draws (Fisher & Yates, 1953). We condition on a reference/context (green) and the current input (blue) to sample values for the masked tokens (white). The model samples a permutation that unscrambles to "forest" on the left, and "fortes" on the right. At any point in generation, the partially-masked sequence corresponds to some valid permutation.

# 1 Introduction

Learning in the space of permutations is a fundamental problem with applications ranging from ranking for recommendation systems (Feng et al., 2021), to combinatorial optimization, learning-to-rank

(Burges, 2010), and data cleaning (Kamassury et al., 2025). Classical probabilistic models for permutations include the Plackett-Luce (Plackett, 1975; Luce et al., 1959) and Mallows (Mallows, 1957) distributions, which can only represent a limited set of probability distributions over permutations (e.g., Plackett-Luce cannot model a delta distribution). These limitations have been addressed in existing literature by considering mixtures Lu & Boutilier (2014), which require expensive variational inference procedures for learning and inference. More recently, several works have proposed methods for learning arbitrary probability distributions over permutations using neural networks, in the framework of diffusion (Zhang et al., 2024) and convex relaxations (Mena et al., 2018) (see Section 2 for an overview).

In this work, we develop models that can represent any probability distribution over permutations and can be trained with conventional deep learning techniques, including any-order masked language modelling (MLM) (Uria et al., 2016; Larochelle & Murray, 2011), and autoregressive next-token-prediction (AR or NTP) (Shannon, 1948). We leverage alternative representations for permutations (beyond the usual inline notation) that form a bijection with the symmetric group, allowing for unconstrained learning. The representations we consider stem from well-established algorithms in the permutation literature, such as factorial indexing (Lehmer codes (Lehmer, 1960)), generating random permutations (Fisher-Yates draws (Fisher & Yates, 1953)), and modelling sub-rankings (Insertion-Vectors (Doignon et al., 2004; Lu & Boutilier, 2014)); which all have varying support for their sequence-elements that are a function of the position in the sequence (Section 3.1).

To trade off compute and expressivity, MLMs have the capability of sampling multiple permutation elements independently with one forward pass through the neural network. Aforementioned representations always produce valid permutations at inference time for any amount of compute spent, even in the fully-factorized case when all tokens are unmasked in a single forward-pass.

Decoding the inline notation of the permutation from the representation is trivial in the case of Lehmer and Fisher-Yates (Kunze et al. (2024a)). In Theorem 4.3 we establish a relationship between a permutation's inverse, and its Lehmer and Insertion-Vector representations, which allows us to develop a fast decoding algorithm for Insertion-Vectors that can be applied in batch, significantly improving inference time compute.

Our methods establishes new state-of-the-art results on the common benchmark of solving jigsaw puzzles (Mena et al., 2018; Zhang et al., 2024), significantly outperforming previous diffusion and convex-relaxation based approaches. However, we also argue this benchmark is inadequate to evaluate learning probability distributions over permutations, as each puzzles contains only one permutation that unscrambles it (i.e., the target distribution is a delta function). We therefore propose two new benchmarks, which require learning non-trivial distributions: learning cyclic permutations (Section 5.2) and re-ranking a set of movies based on observed user preference in the MovieLens dataset (Section 5.3).

In summary, our contributions are four-fold. We:

- (Section 4.2) develop new methods for supervised learning of arbitrary probability distributions over permutations that (1) assign zero probability to invalid permutations; (2) can trade-off expressivity for compute at sampling time, without re-training; (3) can learn non-trivial, fully-factorized distributions; (4) is trained with conventional language modelling techniques with a cross-entropy loss; (5) is extremely fast at sampling time;
- (Section 5.1) establish state-of-the-art on the common benchmark of jigsaw puzzles, significantly outperforming current baselines;
- (Section 5.2 and Section 5.3) define two new benchmarks: learning cyclic permutations and re-ranking based on user preference data, that require learning non-trivial distributions;
- (Theorem 4.3) establish a new relationship between insertion-vectors, inverse permutations, and Lehmer codes that result in an efficient decoding scheme for insertion-vectors.

## 2 RELATED WORK

**Generative models and objectives.** We utilize generative models parametrized by transformers Vaswani et al. (2017), as commonly employed in language modeling. Specifically, we utilize Masked Language Modeling (MLM) and next-token prediction (NTP or AR). The concept of NTP goes back

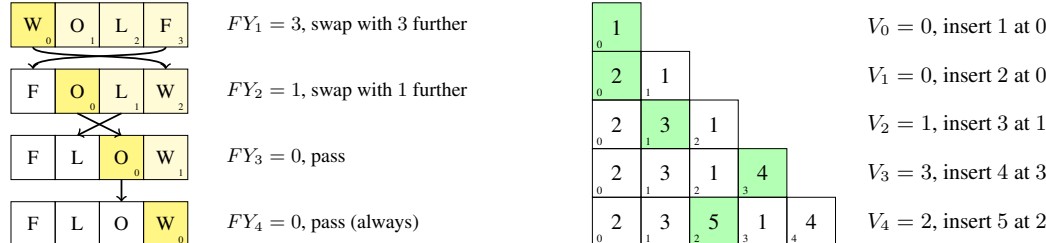

Figure 2: (Left) Illustration for the Fisher-Yates algorithm for shuffling, defining a bijection with permutations. In this example, $FY(X) = [3, 1, 0, 0] \Rightarrow X = [4, 3, 2, 1]$. Small numbers in the bottom-right corner of each box represent the draw value required to swap the current element with that position. (Right) Illustration of the generative process defined by the insertion vector, for a reference permutation $X_{\text{ref}} = [1, 2, 3, 4, 5]$. At each step, the current element of the reference is inserted immediately to the left of $V_i$, and values to the right are shifted right one position to accommodate. Small numbers at the bottom-left corner represent the slot index. In this example, $V(X) = [0, 0, 1, 3, 2] \Rightarrow X = [2, 3, 5, 1, 4]$.

as far as Shannon (1948) and has been applied with great success in language modeling within the last decade, see e.g. Radford et al. (2019); Meta (2024) and many more. Popularized through BERT (Devlin et al., 2019), MLM has been identified as a viable tool for language understanding. More recently, forms of MLM have been derived as a special case of discrete diffusion (Austin et al., 2021; He et al., 2022; Kitouni et al., 2024), where the noise distribution is a delta distribution on the masked state, and have shown promise in generative language modeling (Sahoo et al., 2024; Shi et al., 2025; Nie et al., 2025).

**Permutation and Preference Modeling.** Notable families of distributions over permutations include the Plackett-Luce distribution (Plackett, 1975; Luce et al., 1959) and the (generalized) Mallows model (Mallows, 1957), both of which have restricted expressivity. Doignon et al. (2004); Lu & Boutilier (2014) propose the Repeated Insertion Model (RIM) and a generalized version (GRIM) to learn Mallows models and mixtures thereof, which itself also uses the same insertion representation used in this paper. These methods are detailed in Section 3.2.

A prominent line of related work approaches permutation learning using differentiable ordering. One common strategy is to relax the discrete problem into continuous space—either by relaxing permutation matrices (Grover et al., 2019; Cuturi et al., 2019) or by using differentiable swapping methods (Petersen et al., 2022; Kim et al., 2024). A notable baseline for us is the work of Mena et al. (2018), who utilize the continuous Sinkhorn operator to regress to specific permutations, rather than distributions over possible permutations.

Using Lehmer codes for permutation learning has been considered by Diallo et al. (2020), but only in the AR context and with a different architecture than considered in this work; as well as Malagón et al. (2025) to sample solutions to certain optimization problems in their framework (see "4.2 Case 2: The First-Order Marginal Probabilities Model" in their paper). Recently, Zhang et al. (2024) joined the concepts of discrete space diffusion and differentiable shuffling methods to propose an expressive generative method dubbed SymmetricDiffusers, SymDiff for short. Inspired from random walks on permutations, they identify the riffle shuffle (Gilbert, 1955) as their forward process. To model the reverse process, the paper introduces a generalized version of the Plackett-Luce distribution. This work serves as our most relevant and strongest baseline.

## 3 BACKGROUND

A short introduction to permutations is given in Section A.1.

**Notation** Sequences of random variables are denoted by capital letters $X, L, V$, and $FY$. Subscripts $X_i, L_i, V_i$, and $FY_i$ indicate their elements. Contiguous intervals are denoted by $[n] = [1, 2, \ldots, n]$ and $[n) = [0, 1, \ldots, n-1]$. For some set $S$ with elements $s_j \in [n]$, let $X_S = \{X_{s_1}, \ldots, X_{s_{|S|}}\}$ be the set of elements in $X$ restricted to indices in $S$. For an ordered collection of sets $S_i$, we denote unions as $S_{<i} = \bigcup_{j<i} S_j$. The Lehmer code (Lehmer, 1960), Fisher-Yates (Fisher & Yates, 1953),

and Insertion-vector (Doignon et al., 2004; Lu & Boutilier, 2014) representations of a permutation $X$ will be denoted by $L(X), FY(X)$, and $V(X)$, respectively. We sometimes drop the dependence on $X$ when clear from context or when defining distributions over these representations directly. All logarithms are base 2.

### 3.1 REPRESENTATIONS OF PERMUTATIONS

**Lehmer Codes (Lehmer, 1960).** A Lehmer code is an alternative representation to the inline notation of a permutation. The Lehmer code $L(X)$ of a permutation $X$ on $[n]$ is a sequence of length $n$ that counts the number of inversions at each position in the sequence. Inversions can be counted to the left or right, with one of the following 2 definitions,

$$\text{Left:}\quad L(X)_i = |\{j < i : X_j > X_i\}| \quad \text{or} \quad \text{Right:}\quad L(X)_i = |\{j > i : X_j < X_i\}|. \quad (1)$$

An example of a right-Lehmer code is given in Figure 3. The right-Lehmer code is commonly used to index permutations in the symmetric group, as it is bijective with the factorial number system. The $i$-th element $L(X)_i$ of the right-Lehmer has domain $[n - i + 1)$, and $[i)$ for the left-Lehmer code. A necessary and sufficient condition for a Lehmer code to represent a valid permutation is for its elements to be within their respective domains. The manhattan distance between Lehmer codes relates to the number of transpositions needed to convert between their respective permutations, establishing a metric-space interpretation. This is formalized in Theorem B.1. As a direct consequence, the sum $\sum_i L(X)_i$ equals the number of adjacent transpositions required to recover the identity permutation, known as Kendall's tau distance (Kendall, 1938). Code to convert between inline notation and right- or left-Lehmer codes is given in Section D.1.

**Fisher-Yates Shuffle (Fisher & Yates, 1953).** The Fisher-Yates Shuffle is an algorithm commonly used to generate uniformly distributed permutations. The procedure is illustrated in Figure 2. At each step, the element at the current index is swapped with a randomly selected element to the right, and after $n$ steps is guaranteed to produce a uniformly distributed permutation if the initial sequence is a valid permutation. The index sampled at each step, $FY_i$, are referred to as the "draws". Each resulting permutation $X$ can be produced with exactly 1 unique sequence of draws $FY(X)$, implying the set of possible draw-sequences forms a bijections with the symmetric group (Fisher & Yates, 1953). During the Fisher-Yates shuffle it possible to sample 0, resulting in no swap (see a "pass" step in Figure 2 for an example). If sampling is restricted such that $FY_i > 0$, then the procedure is guaranteed to produce a cyclic permutation and is known as Sattolo's Algorithm (Sattolo, 1986).

Decoding a batch of Fisher-Yates representations can be parallelized by applying the Fisher-Yates shuffle to a batch of identity permutations and forcing the draws to equal elements $FY_i$. Encoding requires inverting the Fisher-Yates shuffle by deducing which sequence of draws resulted in the observed permutation. An algorithm to do so is provided by Kunze et al. (2024b) in Appendix C.1, which can be easily made to work in batch. Code to run Fisher-Yates and Sattolo's algorithm is given in Section D.2.

### 3.2 GENERALIZED REPEATED INSERTION MODEL (DOIGNON ET AL., 2004; LU & BOUTILIER, 2014)

The repeated insertion model (RIM) (Doignon et al., 2004) is a probability distribution over permutations that makes use of an alternative representation to inline, called *insertion-vectors*. The insertion-vector $V(X)$ defines a generative process for $X$, relative to some reference permutation $X_{\text{ref}}$. To generate $X$ given $X_{\text{ref}}$ and $V(X)$, we traverse the reference from left to right and insert the $i$-th element of $X_{\text{ref}}$ at slot $V(X)_i \in [i - 1)$. See Figure 2 for an example.

RIM uses a conditional distribution that is independent of $V_{<i}$ to define the joint over the insertion-vector, i.e., $P_{V_i \mid V_{<i}, X_{\text{ref}}} = P_{V_i \mid X_{\text{ref}}}$, while the Generalized RIM (GRIM) (Lu & Boutilier, 2014) uses a full conditional. GRIM can be used to learn probability distributions over permutations conditioned on an observed sub-permutation. For example, for $n = 4$ and an observed sub-permutation $[2, 1, 4]$, we can set $X_{\text{ref}} = [2, 1, 4, 3]$ such that conditional probabilities $P_{V_4 \mid V_{<4}, X_{\text{ref}}}$ can be learned for all permutations agreeing with the observations, i.e.,

| $V_4 = 0$ | $V_4 = 1$ | $V_4 = 2$ | $V_4 = 3$ |
|---|---|---|---|
| $[\mathbf{3}, 2, 1, 4]$ | $[2, \mathbf{3}, 1, 4]$ | $[2, 1, \mathbf{3}, 4]$ | $[2, 1, 4, \mathbf{3}]$. |

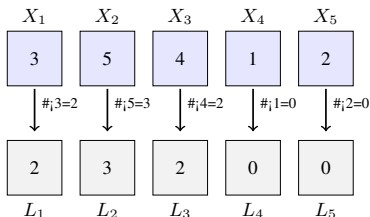

Figure 3: Illustration of the right-Lehmer code for permutation $X = [3, 5, 4, 1, 2]$. (Left) Each $L(X)_i = L_i$ counts the number of elements to the right of $X_i$ that are smaller than it. (Right) Lehmer code interpreted as sampling without replacement indices.

Note this is not possible with inline, Lehmer, or the Fisher-Yates representations. The same can be achieved if the initial elements in $X_{\text{ref}}$ are permuted, as long as the values for $V_{<i}$ are changed accordingly, which highlights an invariance a model over insertion-vectors must learn.

In Lu & Boutilier (2014) the authors use the insertion-vector representation to model user preference data, where the observed sub-permutation represents a partial ranking establishing the preference of some user over a fixed set of items. In Section 5.3 we tackle a similar problem on the MovieLens dataset (Harper & Konstan, 2015) where we rank a set of movies according to observed user ratings.

# 4 LEARNING FACTORIZED DISTRIBUTIONS OVER PERMUTATIONS

This section discusses the main methodological contribution of this work. MLMs can trade off compute and expressivity by sampling multiple permutation elements with one network function evaluation (or forward pass). In that case, simultaneously sampled elements are conditionally independent, which corresponds to an effective loss in modeling capacity. We begin by showing that permutations modeled in the inline representation suffer most from the degradation of model capacity as the number of function evaluations (NFEs) decreases, and can only model delta functions when restricted to a single NFE. We propose learning in the 3 alternative representations discussed in Section 3: Lehmer codes, Fisher-Yates draws, and Insertion-vectors; which do not suffer the same degradation in capacity . Note that while these alternative representations also have constraints for the domain of their elements, these constraints are trivially learned by

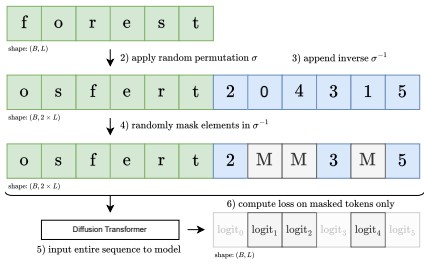

Figure 4: Training our method with MLM during training with the inline notation. For other representations, only the blue tokens change.

the neural network as it only sees valid permutations during training and can infer the domain by setting the appropriate logits to negative infinity. We show the learned conditional distributions defined by these representations are highly interpretable and subsume well known families such as Mallow's model (Mallows, 1957) and RIM (Doignon et al., 2004).

## 4.1 MODELLING CAPACITY OF $P_X^{(\mathcal{S})}$ FOR THE INLINE REPRESENTATION

The masked models considered in this work are of the form,

$$P_X^{(\mathcal{S})} = \prod_i P_{X_{S_i} \mid X_{S_{<i}}} = \prod_i \prod_{j \in S_i} P_{X_j \mid X_{S_{<i}}}, \tag{2}$$

where $\mathcal{S} = (S_1, \ldots, S_k)$ forms a partitioning of $[n]$, and the number of neural function evaluations (NFEs) is equal to $k$. Elements are sampled independently if their indices belong to the same set $S_j$, when conditioned on previous elements $X_{S_{<i}}$. The choice of NFEs restrict $P_X^{(\mathcal{S})}$ to a different family of models through different choices of partitioning $\mathcal{S}$. For example, when limited to 1 NFE, the model is fully-factorized with $S_1 = [n]$. AR minimizes at full NFEs (i.e., $n = k$) with $S_i = \{i\}$, while MLM places a distribution on the partitionings $\mathcal{S}$ resulting in a mixture model.

We consider the problem of *learning distributions over valid permutations* by minimizing the cross-entropy,

$$\min_P \mathbb{E} - \log P_X^{(\mathcal{S})} \text{ subject to } P_X^{(\mathcal{S})}(x) = 0 \text{ if } x \text{ is not a valid permutation,} \tag{3}$$

where the expectation is taken over the data distribution.

Previous works have considered modelling permutations in the inline notation where $X_i$ can take on any value in $[n]$. To produce *only* valid permutations, it is necessary and sufficient for the support of $P_{X_j \mid X_{S_{<i}}}$ to not overlap with that of another index in $S_i \cup S_{<i} = S_{\leq i}$. We can obtain an upper bound on the entropy of *any* inline model by considering the case when all indices in $j' \in S_i$ are deterministic except for some $j \neq j'$, which is uniformly distributed over the remaining candidate indices. Formally, $H(P_{X_{j'} \mid X_{S_{<i}}}) = 0$ and $H(P_{X_j \mid X_{S_{<i}}}) = \log(n - |S_{\leq i}| + 1)$. This implies the following for all $j \in S_i$,

$$H\left(P_X^{(\mathcal{S})}\right) \leq \sum_i \log\left(n - |S_{\leq i}| + 1\right). \tag{4}$$

Equation (4) shows the modelling capacity is severely impacted by the number of NFEs. Most importantly: **any inline model respecting the constraint in Equation (3) can only represent a delta function in the case of 1 NFE** (i.e., $S_1 = [n]$), as $H(P_X^{(\mathcal{S})}) \leq 0$ implies $H(P_X^{(\mathcal{S})}) = 0$ (Cover, 1999). In practice, this manifests at sampling time where the model fails to produce valid permutations as in Section 5.2. At full NFEs the right-hand side of Equation (4) equals $\log(n!)$, and is achievable when $P_X^{(\mathcal{S})}$ is a uniform distribution.

## 4.2 Factorized Representations for Permutations

Next, we consider learning distributions over permutations with the factorized representations discussed in Section 3.1. These representations have different supports for their sequence-elements and allow values to overlap while still producing valid permutations, implying they don't suffer from the representation capacity issue discussed in Section 4.1. At full NFEs, these representations can model arbitrary distributions over permutations, while at a single NFE they can can learn non-trivial distributions such as the Mallow's model and RIM; in contrast to inline which can only represent a delta distribution. For this reason, we refer to them as *factorized representations*.

**Lehmer Codes.** We consider models $P_L^{(\mathcal{S})}$ over the (right) Lehmer code as defined in Section 3.1 and illustrated in Figure 3. Left-to-right unmasking of a Lehmer code can be interpreted as the sampling without replacement (SWOR) indices of its corresponding permutation, as illustrated in Figures 3 and 10. In the AR setting, our model subsumes Mallow's weighted model (Mallows, 1957) over the remaining elements (those that have not yet been sampled).

**Remark 4.1.** The weighted Mallow's model with weights $w_j$ and dispersion coefficient $\phi$ is recovered when $P_{L_j \mid L_{<i}}(\ell_j \mid \ell_{<i}) \propto \phi^{\omega_j \cdot \ell_j}$, for all $j \in S_i$. This follows directly from,

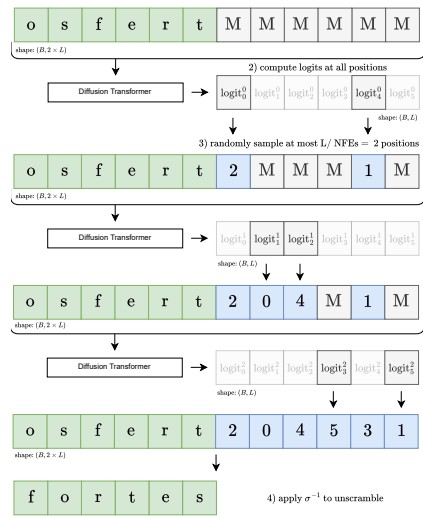

Figure 5: Our method during inference in the inline notation for sequence length $L = 6$ and NFEs = 3. For other representations, only the blue tokens change.

$$P_{L_{S_i} \mid L_{<i}}(\ell_{S_i} \mid \ell_{<i}) = \prod_{j \in S_i} P_{L_j \mid L_{<i}}(\ell_j \mid \ell_{<i}) \propto \phi^{\sum_{j \in S_i} \omega_j \cdot \ell_j}, \tag{5}$$

where $\sum_{j \in S_i} \omega_j \cdot \ell_j$ is the weighted Kendall's tau distance (Kendall, 1938). In particular, when fully-factorized, it can recover the weighted Mallow's model over the full permutation.

**Fisher-Yates.** We define the Fisher-Yates code $FY(X)$ of some permutation $X$ as the sequence of draws of the Fisher-Yates shuffle that produces $X$ starting from the identity permutation. For MLM and AR, unmasking in the Fisher-Yates representation corresponds to applying random transpositions to the inline notation. Similar to Lehmer, this can also be viewed as SWOR, except that the list of remaining elements (faded and bright yellow in Figure 2) is kept contiguous by placing the element at the current pointer (bright yellow in Figure 2) in the gap created from sampling.

**Insertion-Vectors.** We train using the insertion-vector representation to define conditional distributions over sub-permutations. Similar to how Lehmer can recover Mallow's weighted model, conditionals can define a RIM (Doignon et al., 2004) over permutations compatible with the currently observed sub-permutation.

**Remark 4.2.** RIM is subsumed by our model when the insertion probabilities are independent of ordering between currently observed elements, i.e., $P_{V_{S_i} \mid V_{<i}, X_{\text{ref}}} = P_{V_{S_i} \mid X_{\text{ref}}}$.

For Lehmer and Fisher-Yates representations there exist efficient algorithms to convert from (encode) and to (decode) inline, but it is not obvious how to do so for insertion-vectors. The following theorem allows for an efficient batched algorithm for encoding and decoding, by leveraging known algorithms for Lehmer codes (see Section D.1).

**Theorem 4.3.** *Let $L(X)$ be the $k$th element of the left-Lehmer code, $X^{-1}$ the inverse permutation, and $V(X)_k$ the $k$th element of the insertion vector of $X$. Then,*

$$V(X)_k = k - L(X^{-1})_k. \tag{6}$$

The proof follows from the repeated insertion procedure sampling, without replacement, the positions in which to insert values in the permutation. A full proof is given in Section B.2. Code to encode and decode between inline and the insertion-vector representation is given in Section D.3. A more general theorem was proven in Azpeitia et al. (2025)

## 5 EXPERIMENTS

This section discusses experiments with factorized representations, as well as inline, across different losses. We explore 3 experimental settings. First, a common baseline of solving jigsaw puzzles of varying sizes, where the target distribution is a delta function on the permutation that solves the puzzle. We then propose 2 new settings with more complex target distributions: learning a uniform distributions over cyclic permutations, as well as re-ranking movies based on observed user preference. For MLM at low NFEs each set in $\mathcal{S}$ is of size $n/\text{NFEs}$ (rounded), with the exception of the last set. Hyper-parameters for all experiments are given in Section E. An illustration of training is given in Figure 4 and inference in Figure 5.

### 5.1 SOLVING JIGSAW PUZZLES.

We evaluate our models on the common benchmark of CIFAR-10 jigsaw puzzles using the exact same setup as in Zhang et al. (2024). Experimental details are given in Section E. For MLM, we use the same architecture (SymDiff) as Zhang et al. (2024), with the CNN backbone conditioning on the jigsaw tensor. For AR, we modify the architecture to add an additional step that attends to the input sequence as well as the tensor (see Section D.4). All models have roughly 3 million parameters.

Our method significantly outperforms previous diffusion and convex-relaxation baselines, with all representations and losses. Results are shown in Figure 6. MLM can solve the puzzle with 1 NFE (i.e., 1 forward-pass) as the target distribution is a delta on the solution, conditioned on the puzzle.

### 5.2 LEARNING A UNIFORM DISTRIBUTION OVER CYCLIC PERMUTATIONS

The jigsaw experiment is limited in evaluating the complexity of distributions over permutations, as the target is a delta function. In this section we propose a new benchmark where the target distribution is uniform over all $(n-1)!$ cyclic permutations of length $n = 10$.

All cyclic permutations of length $n$ are generated with Sattolo's algorithm (Sattolo, 1986), and a random set of $20\%$ are taken as the training set, resulting in a train set size of $(n-1)!/5$. Results are shown in Figure 7 where each point represents $10,000$ samples. All models learn to fully generalize

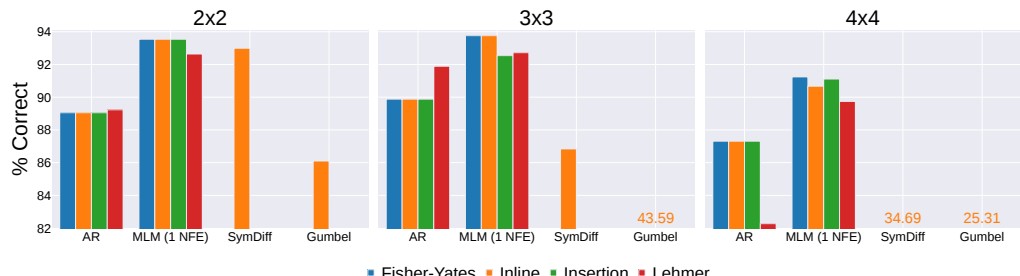

Figure 6: Percentage of CIFAR-10 jigsaw puzzles (test set) correctly reassembled for varying puzzle size, methods, and permutation representation (higher is better). SymDiff (Zhang et al., 2024) and Gumbel-Sinkhorn (Mena et al., 2018) significantly under-perform as puzzle size increases, while our methods do not. Numbers over SymDiff and Gumbel-Sinkhorn indicate their values on the y-axis, which fall below the plotted range. MLM outperforms AR by a wide margin, even while using only 1 NFE.

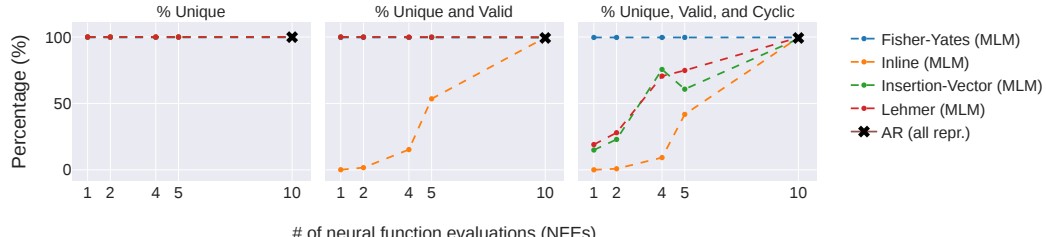

Figure 7: Performance on cyclic generating task as a function of NFEs (i.e., forward passes), across different representations and losses (higher is better). Each point contains information regarding 10k samples. (Left) Percentage of unique output sequences, including invalid permutations. All representations achieve 100%. (Middle) Percentage of simultaneously unique and valid permutations. Except for Inline, all representations achieve 100%. (Right) Percentage of unique, valid, and cyclic permutations. See discussion in Section 5.2.

in the following sense: out of the $10,000$ samples taken, around $20\%$ are in the training set, while the rest are not. All factorized representations can produce valid permutations, even as the number of NFEs decreases, including for the fully-factorized case of 1 NFE. Inline suffers to produce valid permutations as discussed in Section 4.1. All methods can fully model the target distribution at full NFEs, including inline representations (right-most plot). Both Lehmer and Insertion-Vector representations can still produce some cyclic permutations (above the $(n-1)!/n! = 0.1$ baseline) even at 1 NFE. *Fisher-Yates can perfectly model the target distribution for any number of NFEs.* This is expected, as hinted by Sattolo's algorithm: a necessary and sufficient condition to generate cyclic permutations in the Fisher-Yates representation is for $FY_i > 0$, as these represent a pass in the draw. The model produces a uniform distribution over a subset of cyclic permutations. For example, Lehmer at 5 NFEs has non-zero mass on only $46.1\%$ of the $(n-1)!/n!$ cyclic permutations. Within those $46.1\%$, the probabilities are uniformly distributed, while the remaining have 0 mass.

## 5.3 RE-RANKING ON MOVIELENS

Our last experiment is concerned with learning distributions over rankings of size $n$, conditioned on existing user preference data in the MovieLens32M dataset (Harper & Konstan, 2015). MovieLens contains 32 million ratings across $87,585$ movies by $200,948$ users on a $0.5$ scale from $0.5$ to $5.0$. We first filter to keep only movies rated by at least $1,000$ users, and then randomly sample $1,000$ movies from the remaining. Only users that rated at least $n$ movies out of the $1,000$ sampled movies are kept. In the smallest setting ($n = 50$), the dataset totals roughly 18 million ratings across 174 thousand users. The dataset was split on users into $80\%$ train and $20\%$ validation.

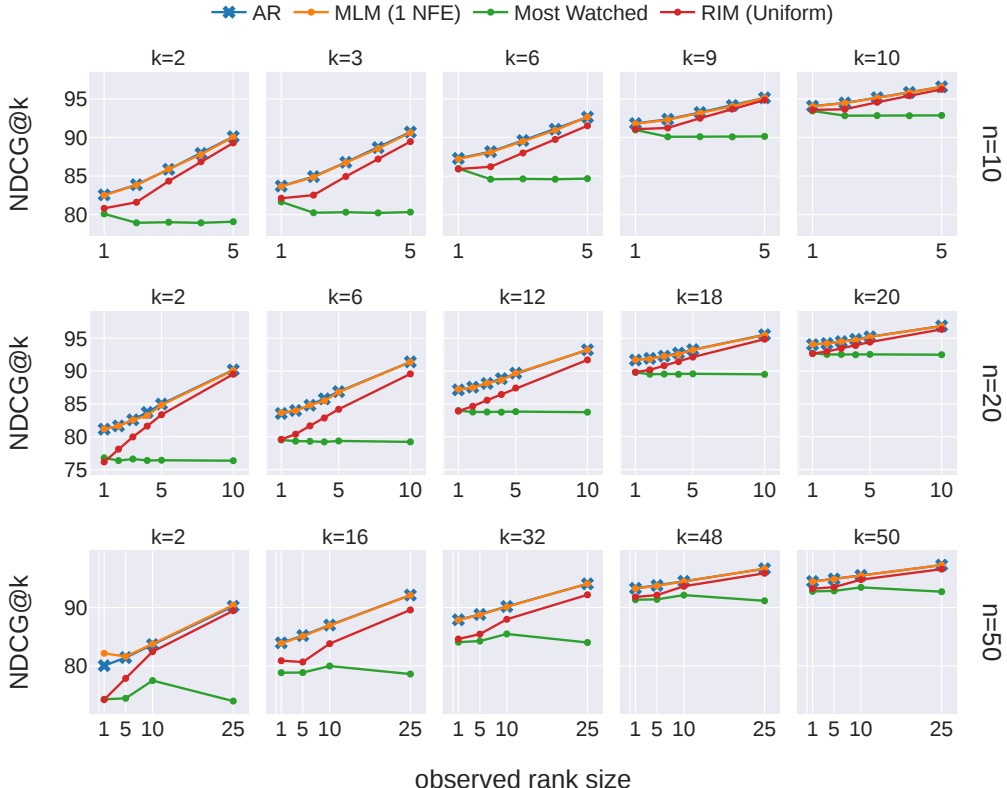

Figure 8: Results for re-ranking conditioned on user ratings in MovieLens (higher is better) for varying rank sizes $n$. See Section 5.3 for a full discussion of the results.

Note that the only information available to the models in this paper are rankings of previous liked items, with no notion of user, user features, or even item features. There is no guarantee of a single "true ranking" of size $n$ when conditioned on a sub-ranking of size $k < n$; as there will likely be two users that have the same preference on a subset of $k$ movies, but differ in preference when looking at the full ranking of size $n$ (i.e., the target is a uniform distribution over these rankings of size $n$).

During training, we sample $n$ ratings (each for a different movie) from each user. The (shuffled) sequence of $n$ movie ids make up $X_{\text{ref}}$. The user ratings are then used to compute the true ranking (i.e., labels), with ties broken randomly. The input sequence is of size $2n$, with the first $n$ corresponding to the movie labels (i.e., $X_{\text{ref}}$, prefix), and the last $n$ the true user ranking in the insertion-vector representation (i.e., $V(X)$, labels). We train with MLM and AR to predict the labels conditioned on the prefix, and the labels generated so far (i.e., conventional cross-entropy training, or "teacher-forcing").

To evaluate, we sample $n$ ratings for each user in the test set (as done in training) and condition on the first few movies $V_{<i}$ to predict the remaining $V_{\geq i}$. Note this is possible without training separate conditional models, because the GRIM representation allows us to learn all conditionals of the form $P_{V_i \mid V_{<i}, X_{\text{ref}}}$ when training with the AR and MLM objectives.

We compare against two baselines: ranking movies by number of users that watched them, and RIM (Doignon et al., 2004) with uniform insertion probabilities; conditioned on the observed ranking $V_{\leq r}$. Results are shown in Figure 8 for the NDCG@k metric (Järvelin & Kekäläinen, 2002). NDCG@k measures the agreement to the true user ratings, and has a maximum value of $1.0$. Note that NDCG@k is similar to cross-entropy when the relevance scores are the normalized log-probabilities (which is our case), which is an appropriate metric for a distribution learning task.

AR ($\prod_{j>r} P_{V_j \mid V_{<j}}$) and MLM (1 NFE, $\prod_{j>r} P_{V_j \mid V_{\leq r}}$) perform similarly, and outperform both baselines in all settings. Note $r = 1$ and $r = 0$ are equivalent, as $V(X)_1 = 0$ with probability 1. The conditional MLM model at 1 NFE is different from the *unconditional* MLM model at 1 NFE

($\prod_{j>r} P_{V_j}$); which is why performance improves as a function of the observed rank size $r$. In this setting, the AR baseline is a very strong baseline, which should have very high performance on this task, given that no semantic content information is available to take advantage of.

## 6 DISCUSSION AND FUTURE WORK

We present models capable of learning arbitrary probability distributions over permutations via alternative representations: Lehmer codes, Fisher-Yates draws, and insertion vectors. These representations enable unconstrained learning and ensure that all outputs are valid permutations. We train our models using auto-regressive and masked language modeling techniques, which allow for a trade-off between computational cost and model expressivity. Our approaches achieve state-of-the-art performance on the jigsaw puzzle benchmark. However, we also argue this benchmark is insufficient to test permutation-distribution modelling as the target is deterministic. Therefore, we introduce two new benchmarks that require learning non-trivial distributions. Lastly, we establish a novel connection between Lehmer codes and insertion vectors to enable parallelized decoding from insertion representations.

The methods in this work explore learning distributions over permutations, where the set of items to be ranked is already known before-hand. An interesting avenue for future work is to model the set of items simultaneously, as is the case in real-world recommender systems. Experiments on MovieLens hint at the scaling capabilities of these factorized representations beyond simple toy settings, as the size of learned permutations for non-trivial experiments in previous literature has generally been much smaller than that explored in our largest MovieLens experiment ($n = 50$). Finally, from a theoretical standpoint there is room for more characterization of the properties of these families of distributions in the low NFE setting.

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

# A BACKGROUND

## A.1 PERMUTATIONS

A permutation in this context is a sequence $X$ of elements $X_i \in [n]$ such that $\bigcup_i \{X_i\} = [n]$ with $X$ having no repeating elements. Permutations are often expressed in inline notation, such as $X = [5, 4, 1, 2, 3]$. A permutation can also be seen as a bijection $X : [n] \to [n]$, where $X(i) = X_i$ is the element in the inline notation at position $i$.

A *transposition* is a permutation that swaps exactly 2 elements, such as $X = [1, 2, 4, 3]$.

A *cycle* of a permutation is the set of values resulting from repeatedly applying the permutation, starting from some value. For the previous example, the cycles are $(1 \to 5 \to 3 \to 1)$ and $(2 \to 4 \to 2)$. A *cyclic permutation* is a permutation that has only 1 cycle, an example is given in Figure 9.

The inverse of $X$, denoted as $X^{-1}$, is the permutation such that $X(X^{-1}(i)) = (X^{-1})(X(i)) = i$.

A *sub-permutation* of a permutation $X$ of length $n$, is a sequence of $m \leq n$ elements $Z_j = X_{i_j}$ that agrees with $X$ in the ordering of its elements, i.e., $i_1 < i_2 < \cdots < i_m$. For example, $[5, 1, 3]$ and $[4, 1, 2]$ are sub-permutations of $[5, 4, 1, 2, 3]$, but $[4, 1, 3, 2]$ is not.

See Marden (2014); Critchlow et al. (1991) for a more complete introduction to permutations and ranking models.

# B THEOREMS AND PROOFS

## B.1 NEIGHBORING LEHMER CODES DIFFER BY A TRANSPOSITION

The following theorem gives a metric-space interpretation for Lehmer codes, and how changes in $L(X)$ affect $X$.

**Theorem B.1.** *For any two permutations $X, X'$, if $\|L(X) - L(X')\|_1 = 1$ then $X$ and $X'$ are equal up to a transposition.*

The proof follows from analyzing the list of remaining elements at each SWOR step, and can be seen from a simple example. Consider the following Lehmer codes $L, L'$ differing only at $L'_3 = L_3 + 1$, their SWOR processes, and their resulting permutations $X, X'$.

| $L_1 = 2$ | 1 | 2 | 3 | 4 | 5 | $X_1 = 3$ |
| $L_2 = 3$ | 1 | 2 | | 4 | 5 | $X_2 = 5$ |
| $L_3 = 1$ | 1 | 2 | | 4 | | $X_3 = 2$ |
| $L_4 = 0$ | 1 | | | 4 | | $X_4 = 1$ |
| $L_5 = 0$ | | | | 4 | | $X_5 = 4$ |

| $L'_1 = 2$ | 1 | 2 | 3 | 4 | 5 | $X'_1 = 3$ |
| $L'_2 = 3$ | 1 | 2 | | 4 | 5 | $X'_2 = 5$ |
| $L'_3 = 2$ | 1 | 2 | | 4 | | $X'_3 = 4$ |
| $L'_4 = 0$ | 1 | 2 | | | | $X'_4 = 1$ |
| $L'_5 = 0$ | | 2 | | | | $X'_5 = 2$ |

Note the following facts:

1. transposing 3 and 1 in the initial permutation (first row) and applying the SWOR process of $L$ results in $X'$;

2. the element chosen at step 3 by $L_3$ is adjacent in the list to the element chosen by $L'_3$, as $|L_3 - L'_3| = 1$;

3. steps before 3 are unaffected, as are their respective inline elements;

4. steps after 3 are unaffected, as long as the sampled index does not fall in either of the two blocks corresponding to $L_3$ and $L_3 + 1$ (where a change occurred).

In general, for an increment at position $j$, the only affected elements are those at $L_j$ and $L_j + 1$, implying $X$ and $X'$ differ exactly by the transposition of these elements.

A more general statement can be given for the case of increments beyond 1. Consider $L'_j = L_j + k$. All future steps $i > j$ with elements $L_i \in [L_i, L_i + k]$ are affected, requiring a permutation of size $k + 1$ to recover $X$.

**Theorem B.2.** *For any two permutations $X, X'$ such that $L(X)_i = L(X')_i$ for all $i \neq j$ then $X$ and $X'$ are equal up to a permutation of $|L(X)_j - L(X')_j| + 1$ elements.*

## B.2 THEOREM 4.3

**Restating Theorem 4.3** *Let $L(X)$ be the kth element of the left-Lehmer code, $X^{-1}$ the inverse permutation, and $V(X)_k$ the kth element of the insertion vector of $X$. Then,*

$$V(X)_k = k - L(X^{-1})_k.$$

First, let $p_k$ be the position of the value $k$ in $X$, i.e. $X_{p_k} = k$. By definition of inversion, $p_k = X_k^{-1}$. Then, note $V(X)_k = |\{j < p_k | X_j < k\}|$. In words: The insertion vector element $V(X)_k$ counts the number of elements to the left of the position of value $k$ in $X$ (i.e. $p_k$) that are smaller than $k$. This can be seen by the following argument: By definition, an insertion vector element $V(X)_k$ describes in which index to insert an element with the current value $k$ (or $k + 1$, depending on indexing definitions), see Figure 2 (right). Because all previously inserted values are smaller than $k$ and all values inserted later will be larger, the index at the time of insertion is equal to the count of smaller elements to the left of the final position of value $k$ in $X$, which is $p_k$.

Recall the definition of the left Lehmer code: $L(X)_k = |\{j < k | X_j > X_k\}|$.

Define $L'(X)_k = k - L(X)_k$ and notice that

$$L'(X)_k = k - L(X)_k = k - |\{j < k | X_j > X_k\}| = |\{j < k | X_j < X_k\}|, \qquad (7)$$

since $|\{j < k\}| = k$ and $X_j \neq X_k \quad \forall j < k$.

Insert the inverse permutation $X^{-1}$:

$$L'(X^{-1})_k = |\{j < k | X_j^{-1} < X_k^{-1}\}| = |\{j < k | p_j < p_k\}|$$

Next, perform a change of variable on $j$ in $V(X)_k$:

$$V(X)_k = |\{j < p_k | X_j < k\}| = |\{p_l < p_k | l < k\}| \qquad \text{where} \quad l = X_j \Leftrightarrow j = p_l$$

Comparing,

$$k - L(X^{-1})_k = L'(X^{-1})_k = |\{j | j < k, p_j < p_k\}| = |\{l | p_l < p_k, l < k\}| = V(X)_k.$$

## C LIMITATIONS

The most important limitation of this work is scalability to large permutations. A loose bound can be estimated by realizing that we model the permutations with transformer architectures. Therefore, the memory and compute required to train on tasks that require large permutations are quadratic. In particular, common methods in ranking include score functions, which can act on each item individually to produce a score, rather than needing to condition on all items as we do.

In general, since the search space of permutations grows much quicker with length ($n!$), the scalability is often not dominated by memory requirements if search is required, rather by the compute needed for the search.

An inherent limitation of the method is that $n$ forward passes through the network are needed to achieve full expressivity over the space of permutations of length $n$. This is a consequence of MLM and AR training, resulting in token-wise factorized conditional distributions. This is detailed in Section 4.1.

## D CODE

### D.1 LEHMER ENCODE AND DECODE

In practice, our left-Lehmer encoding maps an inline permutation to $L'$ from Equation (7), because it interacts more directly with the insertion vector.

```python
def lehmer_encode(perm: Tensor, left: bool = False) -> Tensor:
    lehmer = torch.atleast_2d(perm.clone())
    n = lehmer.size(-1)
    if left:
        for i in reversed(range(1, n)):
            lehmer[:, :i] -= (lehmer[:, [i]] <= lehmer[:, :i]).to(int)
    else:
        for i in range(1, n):
            lehmer[:, i:] -= (lehmer[:, [i - 1]] < lehmer[:, i:]).to(int)

    if len(perm.shape) == 1:
        lehmer = lehmer.squeeze()
    elif len(perm.shape) == 2:
        lehmer = torch.atleast_2d(lehmer)

    return lehmer

def lehmer_decode(lehmer: Tensor, left: bool = False) -> Tensor:
    perm = torch.atleast_2d(lehmer.clone())
    n = perm.size(-1)
    for i in range(1, n):
        if left:
            perm[:, :i] += (perm[:, [i]] <= perm[:, :i]).to(int)
        else:
            j = n - i - 1
            perm[:, j + 1 :] += (perm[:, [j]] <= perm[:, j + 1
                :]).to(int)

    if len(lehmer.shape) == 1:
        perm = perm.squeeze()
    elif len(lehmer.shape) == 2:
        perm = torch.atleast_2d(perm)

    return perm
```

### D.2 FISHER-YATES ENCODE AND DECODE

```python
def fisher_yates_encode(perm: torch.Tensor) -> torch.Tensor:
    original_num_dims = len(perm.shape)
    perm = torch.atleast_2d(perm)
    B, n = perm.shape
    perm_base = torch.arange(n).unsqueeze(0).repeat((B,
        1)).to(perm.device)
    fisher_yates = torch.zeros_like(perm).to(perm.device)
    batch_idx = torch.arange(B).to(perm.device)

    for i in range(n):
        j = torch.nonzero(perm[:, [i]] == perm_base, as_tuple=True)[1]
        fisher_yates[batch_idx, i] = j - i

        idx = torch.stack([torch.full_like(j, i), j], dim=1)
        values = perm_base.gather(1, idx)
        swapped_values = torch.flip(values, [1])
        perm_base.scatter_(1, idx, swapped_values)
```

```
18      if original_num_dims == 1:
19          fisher_yates = fisher_yates.squeeze()
20      elif original_num_dims == 2:
21          fisher_yates = torch.atleast_2d(fisher_yates)
22
23      return fisher_yates
24
25  def fisher_yates_decode(fisher_yates: Tensor) -> Tensor:
26      B, n = fisher_yates.shape
27      perm = torch.arange(n).unsqueeze(0).repeat((B,
            1)).to(fisher_yates.device)
28      batch_idx = torch.arange(B).to(fisher_yates.device)
29      for i in range(n):
30          j = fisher_yates[:, i] + i
31          perm[batch_idx, j], perm[:, i] = perm[:, i], perm[batch_idx, j]
32      return perm
```

### D.3   INSERTION-VECTOR ENCODE AND DECODE

```
1  def invert_perm(perm: Tensor) -> Tensor:
2      return torch.argsort(perm)
3
4  def insertion_vector_encode_torch(perm: Tensor) -> Tensor:
5      inv_perm = invert_perm(perm)
6      insert_v = lehmer_encode_torch(inv_perm, left=True)
7      return insert_v
8
9
10 def insertion_vector_decode_torch(insert_v: Tensor) -> Tensor:
11      inv_perm = lehmer_decode_torch(insert_v, left=True)
12      perm = invert_perm(inv_perm)
13      return perm
```

### D.4   MODIFIED SYMDIFF-AR

We modify the following function in `https://github.com/DSL-Lab/SymmetricDiffusers/blob/6eaf9b33e784e72f8b987cf46c97ff5423b74651/models.py#L357C9-L357C26`.

The first $N$ elements of embd correspond to the embeddings of the puzzle pieces computed with the CNN backbone, while the following $N$ are the token embeddings of the input. The attention mask (embd_attn_mask) guarantees all tokens attend to the puzzle pieces, but the inputs can be attended to causally (if perm_attn_mask is causal, AR case) or fully (MLM).

```
1  def apply_layers_self(
2      self, embd, time_embd, attn_mask=None, perm_attn_mask=None,
            perm_embd=None
3  ):
4      N = embd.size(1)
5      time_embd = time_embd.unsqueeze(-2)
6      embd = embd + time_embd
7
8      embd_attn_mask = None
9      if perm_embd is not None:
10          embd = torch.cat([embd, perm_embd], dim=1)
11          embd = self.perm_pos_encoder(embd)
12
13          if perm_attn_mask is not None:
14              embd_attn_mask = (
15                  torch.zeros((2 * N, 2 *
                        N)).to(bool).to(perm_attn_mask.device)
16              )
```

```
17              embd_attn_mask[:, :N] = True
18              embd_attn_mask[N:, N : 2 * N] = perm_attn_mask
19              embd_attn_mask = ~embd_attn_mask
20
21          for layer in self.encoder_layers:
22              embd = layer(embd, src_mask=embd_attn_mask)
23
24          return embd[:, N : 2 * N]
```

## E    EXPERIMENTS

### E.1    JIGSAW EXPERIMENTS

Each CIFAR-10 image is partitioned into a jigsaw puzzle in grid-like fashion. The pieces are scrambled by applying a permutation sampled uniformly in the symmetric group. This produces a tensor of shape $(B, N^2, H/N, W/N)$, where $B$ is the batch dimension, $N$ the puzzle size (specified per dimension) and $H$ and $W$ are the original image dimensions (i.e. $H = W = 32$ for CIFAR-10). The images are cropped at the edges if $H$ and $W$ are not divisible by $N$, as in Zhang et al. (2024).

Hyperparameters:

1. learning rate = $3 \times 10^{-4}$
2. batch size = $1024$
3. Model configurations follow those in `https:// github.com/DSL-Lab/SymmetricDiffusers/tree/ 6eaf9b33e784e72f8b987cf46c97ff5423b74651/configs/ unscramble-CIFAR10`

### E.2    CYCLIC EXPERIMENTS

1. learning rate = $3 \times 10^{-4}$
2. batch size = $1024$
3. DiT model size:
   (a) hidden dimension size = $128$
   (b) number of transformer heads = $8$
   (c) time embedding dimension = $0$
   (d) dropout = $0.05$
   (e) number of transformer layers = $8$

### E.3    RERANKING MOVIELENS

1. learning rate = $3 \times 10^{-4}$
2. batch size = $1024$
3. DiT model size:
   (a) hidden dimension size = $256$
   (b) number of transformer heads = $8$
   (c) time embedding dimension = $0$
   (d) dropout = $0.05$
   (e) number of transformer layers = $10$

## F    COMPUTE

Our experiments were run on nodes with a single NVidia A-100 GPU. Since the models trained are of small scale, no experiment took longer than 2 days to converge. In total, an estimated 10000 GPU hours were spent for the research for this paper.

## G IMPACT STATEMENT

This paper presents work whose goal is to advance the field of Machine Learning. There are many potential societal consequences of our work, none which we feel must be specifically highlighted here.

## H EXTRA FIGURES

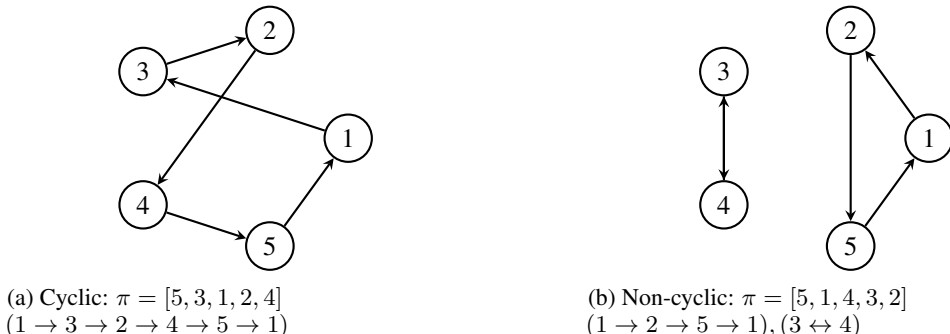

(a) Cyclic: $\pi = [5, 3, 1, 2, 4]$
$(1 \to 3 \to 2 \to 4 \to 5 \to 1)$

(b) Non-cyclic: $\pi = [5, 1, 4, 3, 2]$
$(1 \to 2 \to 5 \to 1), (3 \leftrightarrow 4)$

Figure 9: Illustration of a cyclic vs. a non-cyclic permutation.

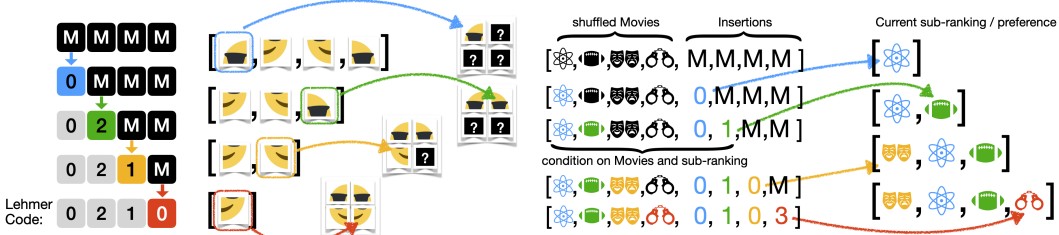

Figure 10: (Left) Decoding a lehmer code from left to right represents sampling without replacement. Illustrated on Jigsaw puzzles. (Right) Prediction task on the MovieLens dataset. Insertion-vectors allow us to define conditionals over sub-rankings corresponding to user preference data.

