# OpenReview forum: "Learning Distributions over Permutations and Rankings with Factorized Representations"
_ICLR.cc/2026/Conference — ICLR 2026 Poster_

### Official Review · Reviewer_ZSYA · 2025-10-23

**Soundness:** 3
**Presentation:** 2
**Contribution:** 2
**Rating:** 4
**Confidence:** 4

**Summary:**

In this paper, the authors approach the problem of learning probability distributions over permutations. Although this task might look trivial, the number of probability models over permutations is limited (Mallows, Plackett-Luce, Bradley-Terry,…), and each of them imposes a certain structure in the learnt distribution (strong unimodality, …). In this sense, given a sample of permutations, learning the probability distribution that best resembles the features of the samples is challenging. In this work, the authors consider the inversion-vector type representation to transform the permutations, and learn the distributions on top of using masked language models.

**Strengths:**

S1. Although the problem dates back 50 years, it is still a relevant and unsolved problem today.

S2. I did not read about Fisher-Yates draws previously, and it was definitely relevant to consider as an alternative representation of permutations.

S3. The paper is very well written.

**Weaknesses:**

W1. Many relevant references are missing. For me, it is surprising not to refer to general literature on this topic as:
—> Marden 1995, Analyzing and Modeling Rank Data.
—> Critchlow et al.1991, Probability models on rankings.
—> Fligner et al.1998, Multistage ranking models.

In fact, Critchlow et al. (1991) analyze four properties of probability models on rankings (strong unimodality, L-decomposability, …) that could be interesting to consider in this paper. What can the authors say about the properties of the distributions learnt?

W2. At times, the authors present the use of alternative representations as if it were novel, which it is not. In fact, Mallows' Generalized model itself uses V_j to decompose permutations (incidentally, I have not seen any mention of this model; I recommend that the authors review Fligner 1998).

W3. Furthermore, there is a group of authors who have worked on the use of alternative representations to propose algorithms (some based on NN+RL) that learn probability distributions to sample solutions to permutation problems. A recent work that uses feed-forward networks to estimate probabilities on the inversion vectors is:
—> Malagon et al. 2024, A Combinatorial Optimization Framework for Probability-Based Algorithms by Means of Generative Models. ACM TELO.
In fact, in this work, it is shown that one of the strong points is the possibility to infer in parallel a great number of permutations. So, contribution 5 in Section 4.2 is not entirely new.

W4. Regarding Theorem 4.3, I have to say that this result has been published too; actually, even a more general outcome is proposed. See Table 1 in the work below:
—> Malagon et al. 2025, Una Visión Unificada de Transformaciones Biyectivas en la Optimización de Problemas de Permutaciones, MAEB2025 (English version accepted for ECAI 2025).
In that work, the authors consider inversion vectors, Lehmer codes, and the RIM as a part of the same framework and use them with a unified notation. So, the published version is even more general than that in the present work.

W5. Several times, the authors mention the idea of unconstrained learning. I think it would help if the authors first refer to the mutual exclusivity constraint that appears naturally in permutations, and then suggest that the concept disappears in the inversion vectors space. However, in some sense, inversion vectors have also "constraints" since the integer numbers that can appear at each position are strict. However, the authors do not mention this.

**Questions:**

Taking into account the comments above:

Q1. What is the novelty of the work? Are you merely proposing to use masked language models (MLM) with alternative representations?

Q2. Can you describe any properties of the masked models learned, following Critchlow 1991?

Q3. Would it be possible to extend the idea of MLM as adapted from Uria 2016 in this paper to solve optimization problems?

If the authors consider the suggested comments and make changes to the paper so that it provides a much more comprehensive overview of the literature and its possibilities, I would be willing to reconsider my score.

---

> ### Author Response · Authors · 2025-11-24
>
> We thank the reviewer for their review.
> Overall, we appreciate their pointers to the literature and have incorporated them into our text, providing a more comprehensive overview of the literature as suggested.
>
> We have uploaded an updated manuscript with changes in blue.
>
> **W1 / Q2**:  Thanks for pointing out these missing references.  We have added Marden 1995, Critchlow 1991, and Fligner 1998  to the paper.
>
> With regards to Critchlow 1991, our understanding is that none of these properties will apply as our neural generative model can produce an **arbitrary distribution** over permutations.  These properties are constraints on the distribution over permutations.  Certain models described within the paper may obey subsets of these properties, insofar as they recover known probability models (e.g., Lehmer at 1 NFE recovers Weighted Mallow’s).
>
> **W2: use of representations as novel**:  Please note we do not claim to be the first to utilize these classical representations in modeling permutations.  We discuss many past models which already leverage these representations in section “2 Related Work”. We did not previously mention Mallow’s Generalized model and have added the reference as suggested to the same section.
>
> **Q1 novelty**:  Our novelty instead comes from combining these representations in conjunction with language modeling in order to provide flexible and powerful distributions over permutations.  As shown in our paper, this formalism naturally contains many simpler models of permutations as special cases, while also providing a framework to learn new models, including for instance, a neural version of GRIM.  Combining classical representations with masked and autoregressive language modeling leads to highly performant models of permutations with controllable compute cost. We also show benchmarks used by current work are inadequate to evaluate learning distributions over permutations, as tasks are solvable with a delta-function; as well as provide completely new benchmarks.
>
> **W3**:  We thank the referee for pointing out this past work and have added a citation to Malagon 2024.  We emphasize that this past research’s focus was on solving optimization problems, whereas our focus is on learning flexible and strong probability distributions over permutations (i.e., just the green block in Figure 1 of https://dl.acm.org/doi/pdf/10.1145/3665650).
>
> **W4**:  Due to language barriers, we were unaware of this concurrent work. The mentioned paper is in Spanish, but we did our best to read it anyway (we did not find an English version publicly available). Please note that our Theorem 4.3 goes beyond proving theoretical claims, and is specifically useful to construct an efficient batched algorithm for encoding and decoding Lehmer codes (allowing us to take advantage of GPU acceleration to sample a vast number of permutations in parallel). We provide an actual implementation in the appendix. We added a reference of this paper to Theorem 4.3.
>
> **W5**: Indeed, inversion vectors have this constraint. However, the main difference is that this constraint is very easily satisfied and learned by the model as it is trained only on valid permutations. This is clearest in the cyclic experiments, where the Fisher-Yates representation learns to output valid permutations (middle plot in Figure 5) even when fully factorized (i.e. 1 NFE). We have added this discussion to the main text.
>
> **Q3**:  It’s conceivable that the distributions over permutations we propose could also be useful for solving optimization problems over permutations.  However, the optimization setting typically has just one optimal permutation as the solution and therefore corresponds to learning a delta function distribution as the target distribution.  Additionally, optimization over permutations is a well-studied problem with specialized algorithms, including methods such as continuous relaxation in the space of permutations. For these simpler optimization situations, modelling complex distributions over permutations using alternative representations may not be required. Note that this was explored in https://arxiv.org/abs/2410.02942 but relies on beam searching for decoding (see Appendix G of their paper), which significantly increases the number of forward passes through the network by orders of magnitude (in some sense defeating the purpose of a generative  approach to optimization over permutations due to overuse of compute).

---

> > ### Comment · Reviewer_ZSYA · 2025-11-25
> >
> > Thank you very much for you answers. Some comments below to your responses:
> > Q1/W3: It is true that some of the works such as Malagon 2023 or 2024 are for the context of permutation problems. However, it is not necessarily true that optimization problems have only one global optimum, and it is even possible that there are several different ones. For this reason, the need to learn very flexible models remains very high. See the implementations of probability models used withing Estimation of Distribution Algorithms.
> >
> > W4: I agree that the use of inversion vectors is highly parallelizable, and in fact Malagon did so in his TELO paper. And I agree that yes, there are language barriers, but today, with the translation techniques we have at our disposal, it should not be very difficult to analyze the paper in Spanish and study the fit of theorem 4.3 of this paper in Table 1.
> >
> > I must say that I am disappointed with the answers, because the authors have not addressed the core of the concerns raised in W2 and W3.

---

> > > ### Author Response · Authors · 2025-11-25
> > >
> > > Thank you for the engagement in the discussion.
> > >
> > > Please see the revised PDF (uploaded here to OpenReview, changes in blue) where we address W2 by emphasizing previous works in the background section.
> > >
> > > Regarding W3, we agree translating is possible (which we did). However note that while searching for concurrent work (back when we were writing our paper) the Spanish work mentioned did not appear (this is what we meant by “language barriers”). In the PDF we have referenced the paper in Spanish directly in the theorem 4.3, stating a more general one was proven.

---

### Official Review · Reviewer_UPQC · 2025-10-24

**Soundness:** 3
**Presentation:** 3
**Contribution:** 3
**Rating:** 6
**Confidence:** 3

**Summary:**

The authors propose three other parameterizations of permutations as opposed to the typical inline notation when modeling distributions over permutations, and perform experiments using generative models to confirm the efficacy of these parameterizations.

**Strengths:**

The core idea seems novel and natural, as there are seemingly many ways to represent permutations with no intuitive reason to prefer the inline notation.  On the synthetic experiments the results are also quite good.

**Weaknesses:**

For this last task in particular, there are some issues.  The measurement is NDCG@k which as I understand it doesn’t require a distribution?  Even if one explicitly wanted a distribution over rankings for the sake of, say, uncertainty quantification, I also feel a somewhat straightforward baseline is missing from this task.  Namely, one could learn a distribution over each individual ranking of a film conditioned on a user, and then implicitly induce a distribution over rankings by sampling scores for each film and sorting them.

**Questions:**

Can the authors offer specific settings where one would like to learn a distribution over permutations in practice, using a MLM or any other generative model?  I feel there could be more motivation for the generative element of the paper.

---

> ### Author Response · Authors · 2025-11-24
>
> We thank the reviewer for their review and are happy that you concluded “the core idea seems novel and natural” and that the “results are also quite good” on synthetic experiments.
>
> We have uploaded an updated manuscript with changes in blue.
>
> We hope to address your remaining questions:
>
> **Stronger baseline**: Indeed there is a vast literature on recommendation systems methods on the MovieLens dataset. However, note that the only information available to the models in this paper are rankings of previous liked items, with no notion of user, user features, or even item features. This is by design: we extracted these rankings from the MovieLens dataset and stripped them of any type of semantic content with the purpose of evaluating the task of modelling full rankings from partially observed sub rankings. In this sense, the AR baseline is a very strong model (note the NDCG@k are very high), which should have very high performance on this task, given that no semantic content information is available to take advantage of. At each step, the AR model does a forward pass to produce the next element of the ranking, resulting in $n$ forward passes. This means it can model arbitrary probability distributions, and is in fact the strongest instance of the model the reviewer suggested (in this simplified setting), as it can update scores as it generates the ranking.  The goal of this experiment was to show that MLM, using only a single forward pass instead of $n$, achieves the same performance as the most powerful model (AR). We agree it would be interesting to scale this to more real-world recommendation systems, where user and item features are taken into account. However, this is beyond the scope of this paper. We’ve added this discussion to the main text.
>
> **MovieLens evaluation**:  Consider the following example. Say there are 2 users with the following ground truth rankings across five movies, where in the second A and D are swapped:
> - A B C D E
> - D B C A E
>
> When conditioning on the sub ranking B C E, there are two possibilities and therefore the target distribution is not a delta function. This type of situation was encouraged to occur in our experiments by our filtering procedure in “5.3 Re-ranking on MovieLens”. We appreciate the author's concern about NDCG@k and realize that the original text needed clarification.  We measured NDCG@k using log probabilities as the relevance score, which makes NDCG similar to cross-entropy in that it's optimized when the model's distribution is correct.  In this sense, the metric already takes into account the distributional aspect of the task. We have clarified this point in the text.
>
> **Motivation for distribution over permutations**:  In addition to the settings mentioned in the introduction, there are additional applications for the generative element of the paper.  First, a particular setting where probability distributions are crucial is in compression and transmission of rankings (this is a known application in rank modulation, see https://ieeexplore.ieee.org/document/5513604 for an intro). Here, the receiver must estimate a probability distribution over incoming symbols (here, symbols would be the permutations themselves). A probabilistic formulation is also crucial under settings that require uncertainty across permutations.  For example, consider the ranking of sports teams in a league based on a sequence of games played between pairs of teams.  A distribution over permutations better represents possible league rankings compatible with limited or noisy measurements of relative ability, than a single ranked list.  Finally, a search or exploration across permutations, perhaps for optimization purposes, can benefit from a flexible way to sample permutations as explored in https://arxiv.org/abs/2410.02942.

---

> > ### Comment · Reviewer_UPQC · 2025-11-25
> >
> > I appreciate the authors updates, I will keep my positive score.

---

### Official Review · Reviewer_Cnmj · 2025-10-29

**Soundness:** 4
**Presentation:** 4
**Contribution:** 3
**Rating:** 8
**Confidence:** 4

**Summary:**

This paper addresses distribution learning over permutations by replacing the usual inline permutation representation with three factorized representations — Lehmer codes, Fisher–Yates draws, and Insertion-Vectors. The key idea is that these representations have element-wise conditional independence, so masked language-modeling (MLM) or autoregressive (AR) training can produce valid permutations with much more expressivity at low numbers of forward passes (NFEs) than inline notation.

**Strengths:**

1) **Clear, well-motivated idea.**
Replacing inline notation with factorized bijections is simple but powerful: it directly addresses invalid outputs in fully-factorized masked models and provides a tradeoff knob between expressivity and compute.
2) **Theoretical insight.**
The paper formally characterizes why inline representations collapse to degenerate distributions under 1 NFE and proves a useful identity (Theorem 4.3) that enables efficient batched insertion-vector decoding.
3) **Remarkable gains on benchmarks.**
Method outperforms diffusion and relaxation baselines on jigsaw tasks; The paper also extend two new benchmarks: the cyclic and MovieLens experiments. Both results further prove that the choice of representation can be used to model beyond the delta distribution and to handle conditional generation tasks via  insertion-vector.

**Weaknesses:**

1. **Scalability to large n.**  The results of training/serving for very large rankings (e.g., thousands of items) remains unknown. Jigsaw and cyclic (n=10) are illustrative but small; MovieLens experiments are more realistic but limited to subrankings (n=50). Behavior at larger scale is untested.

2. **Representation selection requires prior knowledge.**
Choosing the best factorized representation (e.g., Fisher–Yates for cyclic structure) is sensible but introduces an extra design choice and possible brittleness when task priors are unknown.

**Questions:**

1. **Scaling to larger n.**  While the proposed approach shows strong results on moderate-sized permutations (e.g., n ≤ 50), its scalability to larger sets remains unclear. It would be helpful to include preliminary experiments on larger-scale settings (e.g., n = 100 or 200) to assess both accuracy and computational cost. Prior work such as Zhang et al. (2024) has demonstrated scaling on sorting tasks; including similar experiments would make the empirical evaluation more comprehensive.

2. **Representation selection and adaptability.**  The paper currently relies on manual choice of representation (e.g., Fisher–Yates for cyclic structures). It would strengthen the contribution to discuss or explore automatic strategies for selecting or combining representations—such as heuristic rules, task-dependent priors, or meta-learning approaches. Providing such guidance would make the method more practical and easier to adopt across diverse domains.

3. **Clarification on non-delta distributions (MovieLens benchmark).**
The paper argues that most prior benchmarks (jigsaw puzzles) effectively correspond to learning *delta distributions* over permutations, whereas the proposed MovieLens re-ranking benchmark involves learning a *non-delta* target distribution. However, the paper does not clearly explain why the MovieLens setup yields a non-degenerate distribution. Could the authors elaborate on how the stochasticity or user-item sampling procedure induces a genuine distribution rather than a single ground-truth permutation? More details or visualizations of this distributional diversity would help clarify the distinction.

4. **Conditional independence under Lehmer codes.**
The authors mention that Lehmer code representations exhibit conditional independence across indices, which supports efficient factorization. However, this property seems to hold only when the target permutation distribution is a delta (i.e., deterministic). For non-delta or multimodal permutation distributions, the dependencies across indices should reappear. Could the authors clarify under what assumptions the conditional independence claim holds, and whether it still approximately holds for stochastic targets such as those in the MovieLens setting?

---

> ### Author Response · Authors · 2025-11-24
>
> We thank the reviewer for their comments and recognition of the paper’s strengths.
>
> We have uploaded an updated manuscript with changes in blue.
>
> We now address their questions:
>
> **Scalability to large n**:  We agree this would be the next step: to investigate scalability to larger permutations. We did run experiments for n=200 on the MovieLens dataset and found no issues with scaling, beyond that common to any other transformer-based models (the scaling of attention with context-length). Please note that at inference time, Zhang 2024 relies on beam searching for decoding (see Appendix G of their paper), which significantly increases the number of forward passes through the network. We show significantly better performance can be achieved with *orders of magnitude* less compute (measured in forward passes using the same architecture), and highlight the extreme case where only 1 forward pass is allowed.
>
> **Representation selection**:  We agree the choice of representation is an additional design choice for modeling distributions over permutations. The choice of representation can be guided by at least two factors: (1) prior knowledge of the target distribution (e.g., cyclic experiment) and (2) the nature of available conditioning information in conditional generation tasks (e.g., MovieLens experiments). As an example of (1), in the cyclic experiments the target is a uniform distribution, which belongs to the family defined by the Fisher-Yates representation for any NFE. As an example of (2): say we have the same setting as the jigsaw puzzle experiments, but with some pieces of the puzzle previously filled in (i.e., "the 17th puzzle piece has been placed in the 2nd slot"). This can be conditioned on in the inline representation by setting the 17th element of the response vector to 2 (or 1, if 0-indexed). Similarly, conditioning on user sub-rankings in MovieLens is easily done by setting the initial indices to the observed items (in any order). In the absence of prior knowledge about a given task, we recommend choosing a representation empirically.   Due to the flexibility of modern language modeling, it’s straightforward to simply experiment with different representations and choose the one that works best for the given task and dataset.
>
> **Clarification on non-delta distributions in MovieLens**:  Consider the following example. Say there are 2 users with the following ground truth rankings across five movies, where in the second A and D are swapped:
> - A B C D E
> - D B C A E
>
> When conditioning on the sub ranking B C E, there are two possibilities and therefore the target distribution is not a delta function. This type of situation was encouraged to occur in our experiments by our filtering procedure in “5.3 Re-ranking on MovieLens”. Note that NDCG@k is similar to cross-entropy when the relevance scores are the normalized log-probabilities (which is our case). In this sense, the metric already takes into account the distributional aspect of the task. We’ve added this clarification to the main text in the same section.
>
> **Conditional independence under Lehmer codes**: The Lehmer code representation can represent non-delta distribution even when fully factorized (i.e., 1 NFE). In fact, for 1 NFE the Lehmer code representation is exactly equivalent to the weighted Mallow’s model (as shown in section 4.2). The inline notation is the one that can only represent delta-functions at full NFEs.

---

> > ### Comment · Reviewer_Cnmj · 2025-11-25
> >
> > The authors address my questions, and I will maintain my score.

---

### Official Review · Reviewer_RZJL · 2025-10-31

**Soundness:** 3
**Presentation:** 3
**Contribution:** 3
**Rating:** 4
**Confidence:** 4

**Summary:**

Defining distributions over permutations is a useful machine learning primitive for applications that involve ranking, assigning, etc.

This paper provides new, interesting ways to represent distributions over permutations by leveraging a variety of classic methods for encoding permutations (e.g., using swaps, insertions, etc). Each of these can be coupled with a language model (masked, autoregressive) for defining a distribution over permutations.

The representational capacity of various representations is discussed and experiments on both synthetic and real data are used to compare performance.

**Strengths:**

The paper invokes a number of interesting, classical formulations of permutations that I was not familiar with. I learned a lot from reading the paper.

The paper is a good example of how to combine classic algorithmic formulations with neural networks to get novel/flexible distributions over structured objects.

**Weaknesses:**

I found the  (num function evaluations) NFE formalism confusing. This is used to provide a tradeoff between compute cost and expressivity of the resulting distribution. I understand the NFE=1 and NFE=k regimes, but I don't know how to interpret a value in between and I don't understand how the partitioning was chosen. I also don't know why this variable was so central in the experiments. To me, I was most interested in the different representations, not sweeping over values of NFE.

I have some questions about the experiments. See below.

**Questions:**

I found the MLM vs. AR exposition confusing. It took me a while to realize that things are sampled independently within a masked region. It may have been helpful to define MLM

I found the setup for experiment 5.2 (cylic permutations) confusing. In what sense are you assessing whether the uniform distribution was learned? Can't these models be used to assess the likelihood of a given permutation? Could you check that the likelihood scores are approximately uniform (suggesting that the distribution is approximately uniform)?

In sec 5.3 (MovieLens), why were only certain representation approaches used? I would have expected to see a comparison of all the approaches in Figs 5 and 6. Also, MovieLens has been used for benchmarking a wide variety of learning-to-rank models. I don't expect you to necessarily get SOTA performance, but it would have been useful to understand roughly how well your methods compare.

---

> ### Author Response · Authors · 2025-11-24
>
> We thank the referee for their review and are delighted that you “learned a lot from reading the paper”.
> We hope we are able to address your concerns and questions about the paper.
>
> We have uploaded an updated manuscript with changes in blue.
>
> **NFE formalism confusing**:  We provide clarification on the formalism here.
> - *Interpretation of intermediate values and how partitioning was chosen*:  When the number of NFE’s is below the sequence length, we can choose to partition the variables arbitrarily.  This is an additional degree of freedom, which has been explored in the discrete diffusion literature, where different amounts of decoding can be performed at each step per neural function evaluation.  In our paper, if the sequence length was $L$, we sample $L/NFEs$ new variables in our representation per step, corresponding to the next $L/NFEs$ data elements ordered by the identity permutation. We have added a new figure showing this sampling procedure in Figure 5.
> - *Centrality of NFE to experiments*:  NFE was a focus throughout the paper because it provides a knob to examine how well different representations can perform under varying compute.  We show in theory that the alternative representations we introduce for modeling permutations capture more flexible distributions over permutations than the traditional inline notation with less compute, and our experiments confirm that this theoretical benefit occurs in practice.
>
> **MLM vs. AR exposition**:  We have added a figure 4 that details training as well as figure 5 for inference for these models.
>
> **Cyclic permutations setup**: Yes, the likelihoods are uniform over the cyclic permutations produced by the model. For example, Lehmer at NFEs=5 has non-zero mass on only 46.1% of the $(n-1)!/n!$ cyclic permutations. Within those 46.1%, the probabilities are uniformly distributed. In more detail: the target distribution is defined on all $n!$ permutations, with uniformly distributed mass on cyclic permutations ($(n-1)!/n! = 10\\%$ of the domain) and $0$ mass on the remaining. For all models at full NFEs=10, the KL is almost exactly zero (right-most points in the plot). Note that one natural metric to check would be the KL divergence of the model with respect to the target. However, this quantity is not appropriate in these settings as the model assigning zero probability to any single cyclic permutation is enough for the KL to be infinite. This happens for all models with NFE<10, except Fisher-Yates and AR.
>
> **MovieLens: choice of representations**:  This was discussed briefly in “3.2 Generalized Repeated Insertion Model”, which we detail next. This experiment requires modelling distributions over rankings (i.e., permutations) given sub-rankings, which is only tractable in the insertion-vector representation. For example, say the user has the following preference data for 3 items (B, C, D) out of a universe of 4 (A, B, C, D): BDC (i.e., user prefers B over D and C, as well as D over C). We want a distribution over rankings of (A, B, C, D) given the observed ranking, i.e., we want to compute the following probabilities from the model (assuming the next item is A):
> - $P(ABDC) = P(V_4=0 | BDC) \cdot P(BDC)$
> - $P(BADC) = P(V_4=1 | BDC) \cdot P(BDC)$
> - $P(BDAC) = P(V_4=2 | BDC) \cdot P(BDC)$
> - $P(BDCA) = P(V_4=3 | BDC) \cdot P(BDC)$
>
> Note that $P(V_4=i | BEC)$ is exactly the i-th softmax output of the model. In contrast for inline, one would need to sum over all permutations where (B, C, D) appear in the correct relative ordering (BDC), which quickly becomes intractable.
>
> **MovieLens: baselines**: Indeed there is a vast literature on recommendation systems methods on the MovieLens dataset. However, note that the only information available to the models in this paper are rankings of previous liked items, with no notion of user, user features, or even item features. This is by design: we extracted these rankings from the MovieLens dataset and stripped them of any type of semantic content with the purpose of evaluating the task of modelling full rankings from partially observed sub rankings. In this sense, the AR baseline is a very strong model (note the NDCG@k are very high), which should have very high performance on this task, given that no semantic content information is available to take advantage of. At each step, the AR model does a forward pass to produce the next element of the ranking, resulting in $n$ forward passes. This means it can model arbitrary probability distributions. The goal of this experiment was to show that MLM, using only a single forward pass instead of $n$, achieves the same performance as the most powerful model (AR). We agree it would be interesting to scale this to more real-world recommendation systems, where user and item features are taken into account. However, this is beyond the scope of this paper. We’ve added this discussion to the main text.

---

> > ### Comment · Reviewer_RZJL · 2025-11-25
> >
> > Thank you for comprehensively responding to my questions. I have raised my score to Accept.

---

### Author Response · Authors · 2025-11-28
**Summary for the ACs**

We'd like to summarize the rebuttal process for the new AC.
- The scores after the rebuttal were **8, 8, 6, 6** (avg 7.0) w/ the same confidence scores as now.
- Please note all score increases happened **before** the leakage was announced.
- Reviewer RZJL (4 -> 8) mentions they'd increase their score here: https://openreview.net/forum?id=aE1VU6Ui4M&noteId=wJRUHv8avH
- Reviewer ZSYA (4 -> 6) did not explicitly mention their increase.

All concerns were addressed in the revised version of the PDF (changes in blue).

Thank you.

---

### Meta-Review · Area_Chair_3XJD · 2026-01-09

**Summary:**

The authors develop new methods for learning distributions over permutations and rankings, using (small)
language models. The initial reviews for the paper were mostly borderline (8, 6, 4, 4). The primary
concerns of the reviewers seemed to relate to novelty (ZSYA, in particular, was worried about novelty
compared to a variety of recent and older work on permutation modeling), clarity (RZJL noted they found
various parts of the paper confusing) and empirical evaluation (e.g. one reviewer noted that they were
concerned about baselines and the use of NDCG). Overall, the reviewers didn't seem to have glaring or
"deal-breaking" issues with the paper and found it novel, interesting and one reviewer noted that they
learned a lot from it. It seems that after the discussion multiple reviewers would have (and did) revised their scores up significantly (to at least 8, 8, 6, 4).  Thus the recommendation is to accept the paper.

**Reviewer Concerns:**

It sounds like points of confusion for reviewer RZJL were clarified and addressed. They seemed to indicate
they would raise their score to accept (8).

As far as I can tell from the discussion, reviewer ZSYA was not satisfied with the author response, which
added discussion of related work ("I must say that I am disappointed with the answers, because the
authors have not addressed the core of the concerns raised in W2 and W3").

Reviewer UPQC appreciated the author response but indicated that they would not change their score.

**Reviewer Scores:**

I think reviewer RZJL raised to 8.

Reviewer ZSYA stated they were not satisfied with the author response, which suggests they would not
change their score. However, the authors suggest that the reviewer did in fact change their score. It's
unclear to me what the discrepancy here is.

Reviewer UPQC appreciated the author response but indicated that they would not change their score.
Thus it seems like the scores after the rebuttal period would end up (8, 8, 6 , 4).

---

### Decision · Program_Chairs · 2026-01-26

Accept (Poster)